# Emotional distress and well-being among people with motor neurone disease (MND) and their family caregivers: a qualitative interview study

Cathryn Pinto [1], Adam W A Geraghty [2], Lucy Yardley,[1,3] Laura Dennison [1]

¹Department of Psychology, University of Southampton, Southampton, UK
²Primary Care and Population Sciences, University of Southampton, Southampton, Hampshire, UK
³School of Health Sciences, University of Bristol, Bristol, UK

**Correspondence to**
Cathryn Pinto;
C.L.Pinto@soton.ac.uk

## ABSTRACT

**Objective** We aimed to get an in-depth understanding of the emotions experienced by people with motor neurone disease (MND) and their caregivers, and to explore what impacts emotional distress and well-being.

**Design** Qualitative study using semi-structured interviews with people with MND and caregivers.

**Setting** Participants were recruited from across the UK and took part in interviews conducted either face to face, by telephone or email to accommodate for varying levels of disability.

**Participants** 25 people with MND and 10 caregivers took part. Participants were purposively sampled based on their MND diagnosis, symptoms and time since diagnosis.

**Data analysis** Data were analysed using inductive reflexive thematic analysis.

**Results** Eight broad themes were generated (20 subthemes). Participants described the emotional distress of losing physical function and having a threatened future because of poor prognosis. Keeping up with constant changes in symptoms and feeling unsupported by the healthcare system added to emotional distress. Finding hope and positivity, exerting some control, being kinder to oneself and experiencing support from others were helpful strategies for emotional well-being.

**Conclusion** The study provides a broad understanding of what impacts emotional distress and well-being and discusses implications for psychological interventions for people with MND and caregivers. Any communication and support provided for people with MND and their caregivers, needs to pay attention to concepts of hope, control and compassion.

## STRENGTHS AND LIMITATIONS OF THIS STUDY

⇒ This study provides an in-depth and inclusive account of what impacts emotional distress and well-being using interviews with people with motor neurone disease (MND) and their family members.

⇒ Purposive sampling and flexible recruitment strategies used to capture experiences of people with MND with a range of symptoms, including written interviews with those who had difficulties with speech.

⇒ Participants included people who had been recently diagnosed as well as those who had MND for years, and therefore, captured experiences of emotional distress and well-being at different stages of having MND.

⇒ Fewer people with mild cognitive impairment took part in the study.

⇒ Study design did not allow us to see changes in emotional distress and well-being over time, as symptoms change and people find new ways to adjust to changes.

## INTRODUCTION

Motor neurone disease (MND) is a neurodegenerative disease, which results in declining physical function and has a very poor prognosis.[1] In terms of psychological impact, many people with MND (pwMND) experience depression.[2–4] Anxiety is prevalent around the time of diagnosis[5 6] and during the final stages.[7] PwMND also experience feelings of hopelessness[8–10] and demoralisation.[2]

Research has described the impact of the significant losses that come with MND on personal, social and occupational relationships.[10] Family members or caregivers of pwMND often struggle with the emotional impact and burden of the disease, and have high rates of psychological morbidity.[11–13] Studies with caregivers have described the strain from caring tasks, from having to make changes to their own lives and not having time for themselves.[14–16] Distress in MND has largely been described as anxiety and depression,[8 12 13 17] or caregiver burden and strain.[18 19] Well-being has largely been described as reduced anxiety and depression or improvements in quality of life.[20–22] In this paper, we use the terms 'emotional distress' to refer to the broad range of negative or difficult thoughts and emotions, and 'emotional well-being' for the broad range of positive or helpful thoughts and emotions experienced by pwMND and caregivers.

Studies have looked at demographic and clinical factors that might explain emotional distress or protect against it.[22] Results from these studies show that psychosocial factors like coping strategies and social support, are more strongly related to well-being and quality of life than demographic and clinical factors.[18 22 23] Emotions are also affected by low self-esteem,[4 18] end-of-life concerns,[8] faith/existential concerns,[24 25] sense of loss[25 26] and changes in identity, roles and relationships.[18 25 27] Psychosocial factors are clearly important for distress and well-being in MND, and we need a better understanding of what influences emotional distress and the use of different coping strategies.[20 28]

Qualitative research can be useful for understanding emotional distress and well-being because it allows participants to express their own understandings and experiences without being limited to concepts predetermined by the researcher. Qualitative studies have largely explored the experience and impact of living with MND,[10 14 25 29] which have added to our understanding of distress in MND. Few studies have focused specifically on the emotions experienced. One qualitative study examined the use of metaphors to express emotions,[30] and a small number have focused on emotions at specific time periods (eg, during diagnosis or the final stages).[6 26 31] Emotional distress at other time points and things that influence distress and well-being need further exploration.

This study aims to build on our understanding of emotional distress and well-being in MND. More importantly, we aim to hear from people whose experiences are not well-represented in research, including people with speech difficulties, cognitive impairment and people at different stages of the disease. It is particularly timely to do this research because recent articles have highlighted that there are few effective psychological interventions to improve well-being,[32] and more research and interventions are needed.[12 33] This study is part of a larger project to develop an intervention to improve emotional well-being among pwMND and caregivers.

In this current study, we aim to understand the emotional impact of living with MND and explore what impacts experiences of emotional distress and well-being among pwMND and their family caregivers.

## METHODS
### Design
Qualitative study using in-depth semi-structured interviews and reflexive thematic analysis, in line with an interpretivist approach. The paper is reported in accordance with the Consolidated criteria for reporting qualitative research (see online supplemental file 1).[34]

### Participants
We aimed to recruit 20–30 pwMND and used purposive sampling to represent people with difficulties with movement, speech and cognition, and different lengths of time since diagnosis. Caregiver participants had fewer sampling criteria (age, gender), therefore, we aimed to recruit 10–15 caregivers.

Eligibility: Participants were above 18 years of age, had an MND diagnosis and had mental capacity to consider participation in the study (assessed by the researcher through correspondence about the study). We included participants who self-reported difficulties with cognition, but had mental capacity to give informed consent, as the views of pwMND who have cognitive impairment have been underrepresented in previous research. We included caregivers above 18 years of age, both current and recently bereaved (bereavement within 1 year from the time of the interview).

### Data collection
Participants were recruited through a UK charity that supports pwMND and their families (MND association). Study information was circulated via the charity's website, newsletters, social media outlets and local support groups. People willing to take part contacted CP, who screened for eligibility and provided further details about the interview.

Before each interview, participants gave written informed consent and filled a demographic/clinical details form. This process was completed either in-person, by post or email, based on the interview mode. CP, who has training and experience in conducting interviews for qualitative research, conducted all interviews. Interview mode was flexible (face to face, email or phone) to accommodate for various levels of disability. Face-to-face interviews were conducted in a place convenient to participants, usually at home or a hospice. Where two members of a couple had both consented to participate, participants were interviewed separately where possible, but jointly where this was requested for reasons of comfort or to facilitate easier communication. The interview topic guide was developed iteratively by CP, LD, AG and patient and public involvement (PPI) members. In line with an interpretivist approach, questions were broad and open ended to allow participants to give rich, in-depth accounts of their emotional experiences in relation to MND, and follow-up questions were led by participants' responses. The final interview topic guide (online supplemental file 2) covered questions about people's experiences living with MND, with a focus on their thoughts and feelings and coping with emotional concerns.

### Data analysis
Face-to-face and phone interviews were audio recorded and transcribed, and all interviews were anonymised. Field notes were completed after each interview, reflecting on participants' responses and interview method and procedure. Data were analysed using reflexive thematic analysis,[35 36] as this was a flexible method that suited the research question, helped us identify common patterns across participants' experiences and allowed us to look for underlying meaning behind experiences of emotional distress and well-being. In line with an interpretivist

approach, we used an inductive approach to data analysis, and included convergent and divergent cases in theme development. The analysis was mainly conducted by CP, a PhD student with previous experience as a qualitative researcher. After familiarisation with the interview transcripts, NVivo V.12 software was used to code the data inductively, focusing on semantic and latent features of the data. The codes went through several iterations as new interviews were coded. During the coding process, notes were made about interesting features of the data and how different concepts related to each other. Following this, similar codes about emotional distress and emotional well-being were grouped together and candidate themes were generated. Themes were then reviewed to see if they represented experiences across the dataset. Theme names were revised and findings written up; this was also an iterative process whereby descriptions were clarified and overlap between themes identified. CP had regular meetings with qualitative research experts LD and AG who helped refine codes and themes. A lay summary of the findings was sent to all participants.

Written informed consent was obtained for all participants. After the interview, participants were debriefed and steps were taken to ensure participants were not distressed.

## Patient and public involvement
Three PPI members (one person with MND and two former caregivers) contributed to this study. They were involved in the recruitment stage to help identify potential participants. They also helped pilot the interview topic guide, refine interview questions and trial the data collection procedure to ensure that it was not burdensome. Some PPI members looked over early drafts of the findings and offered insight about elements to highlight or discuss in the reporting of the results.

## RESULTS
Participants were 25 pwMND and 10 family caregivers (see table 1).

Twenty-nine interviews were conducted in total; 6 joint interviews and 23 one-to-one interviews with either pwMND or caregiver. Fourteen interviews were conducted face to face, eight via telephone and seven via email. Interviews lasted an average of 39.6 min.

We developed 8 themes and 20 subthemes. The first four themes relate to triggers of emotional distress (figure 1); the remaining four themes capture strategies to improve emotional well-being (figure 2). All themes were present in some degree and form in both patient and caregiver interviews; where differences between these groups were apparent, they are described.

## Triggers of emotional distress
Participants described four main triggers of emotional distress and how they had an impact on emotions (themes are numbered, subthemes in italics).

**Table 1** Demographic and clinical details of the sample

| Characteristic | People with MND (n=25) | Caregivers (n=10) |
|---|---|---|
| Age (range 39–80) | | |
| 35–50 years | 3 | 0 |
| 51–65 years | 12 | 4 |
| 66–80 years | 10 | 6 |
| Gender | | |
| Male | 15 | 5 |
| Female | 10 | 5 |
| Relationship to person with MND | | |
| Spouse/partner | | 10 |
| Diagnosis | | |
| ALS limb | 18 | |
| ALS bulbar | 1 | |
| Primary lateral sclerosis | 4 | |
| Progressive bulbar palsy | 1 | |
| Progressive muscular atrophy | 1 | |
| Time since diagnosis (range 2 months to 17 years) | | |
| Less than 1 year | 8 | |
| 1–3 years | 6 | |
| 3–5 years | 7 | |
| More than 5 years | 4 | |
| Difficulties reported often/always* (ALSAQ-5 scale) | | |
| Difficult to stand up | 16 | |
| Difficult to use arms and hands | 16 | |
| Difficulty eating solid food | 4 | |
| Speech is not easy to understand | 9 | |
| Feeling hopeless about the future | 3 | |
| Self-reported concerns about cognitive ability | 2 | |

*ALSAQ-5 is a patient self-report questionnaire used to briefly measure the impact of ALS/MND on patients.[51]
ALS, Amyotrophic Lateral Sclerosis; ALSAQ-5, Amyotrophic Lateral Sclerosis Assessment Questionnaire 5; MND, motor neuron disease.

## Losing function or ability
Most participants described how losing function or ability was distressing because of a *reduced sense of autonomy and control*. PwMND spoke about feelings of anger, frustration, sadness or grief at not being able to do the things they wanted to, at losing their independence and relying on others for help. Caregivers expressed similar emotions at seeing their family member lose capability and independence.

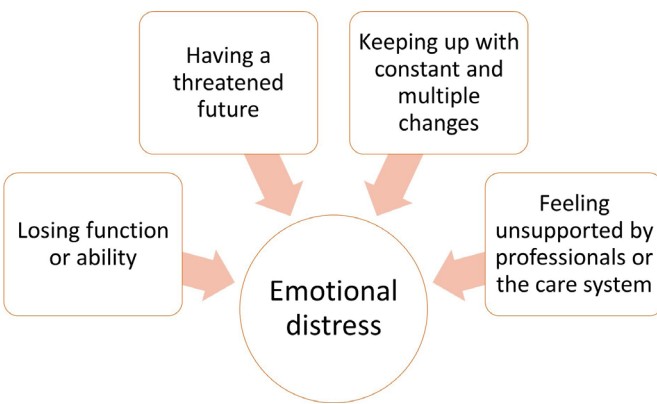

**Figure 1** Triggers of emotional distress.

Hobbies that I had, I can't pursue any of them because I just don't have any capability enough in my arms now to do that. So, I think mentally I've seen a dip in the last six months into a much more kind of negative and angry stage really. (Person with MND, P13)

…and seeing him, and it saddened me, where you're sort of doing the job of hoisting and moving, the manual handling, and then you step back and you're seeing it from a distance, that's actually quite [pauses] hard as well, because he's got no control over his body and he's having to have people do that for him. (Caregiver, C05)

Losing function led to *changes in self-identity*, specifically those of becoming a 'patient' and 'carer'. These changes, especially losing one's voice which is strongly linked to identity, were experienced as distressing.

Losing mobility gradually was bad enough… but losing my voice is much more dreadful. It's taking away part of my personality. I can no longer express myself properly. (Person with MND, P21)

Losing function or ability sometimes led to *changes in relationships and interactions*. PwMND spoke about negative feelings of becoming a burden for their partner/spouse. Both patient and caregiver participants spoke about how

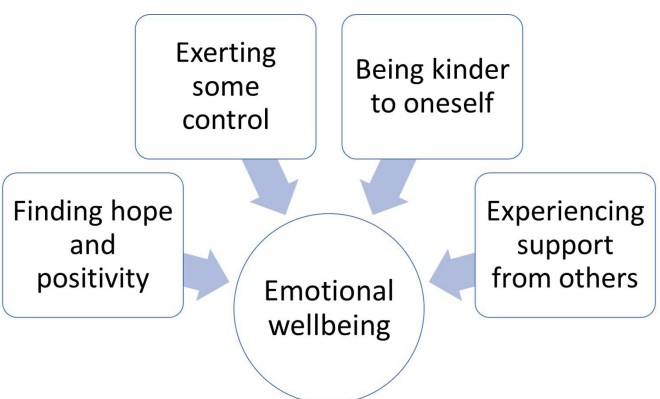

**Figure 2** Strategies used to improve emotional well-being.

having MND sometimes changed or limited interactions between family members.

It's emotionally difficult not being able to physically help my son or my parents and siblings and friends. Not being able to hold and play with my nephew. (Person with MND, P26)

Participants also had *practical concerns or worries about managing tasks* as a result of losing more ability and function.

My main worry at the moment really is moving him from there to there because his knees and lower legs are getting weaker and I've got that (points to hoist) to move him. (Caregiver, C03)

### Having a threatened future
The short prognosis that accompanies a diagnosis of MND led to feelings of anger, disappointment, sadness and feelings of being robbed/cheated of a future (*threat to life narrative and future plans*).

But I'm having to accept I might have a lonely old age. Disappointment, more than anything else. (Caregiver, C27)

Participants also described emotional distress when *anticipating future symptoms or disease progression*. Many things triggered worry about the future including researching MND online, meeting other pwMND, discussions with healthcare professionals and going through symptom checklists. Some participants also spoke about experiencing such worries about the future even if symptoms had not progressed or were relatively stable. This was true particularly just after diagnosis, where worries about the future led to feelings of depression, low mood and withdrawal from others.

Even though physically, say that first month afterwards, there was almost no change but my mind's thinking quite morose almost. You almost think of death at that point… It's not even anger at that stage, it's just hopelessness at that point because you're just thinking oh that is it isn't it? It's all over. (Person with MND, P10)

The *uncertainty of the disease progression* also caused emotional distress among both patients and caregivers, because of being unable to manage symptoms or plan for the future.

I think the biggest thing about this is that it's such an unknown. Because everybody is different and, who knows how much, how long this disease is going to affect you. And the uncertainty, for me, it's very hard to cope with. (Caregiver, C15)

### Keeping up with constant and multiple changes
Participants discussed how the *timing and relentless nature of changes* in symptoms was difficult physically and

emotionally. As time went on, for some it became easier to fall into a care routine. However, if deterioration happened quickly, both patient and caregivers expressed that it was difficult to cope and left them feeling tired, not in control and in need of respite.

> If something's difficult, we find a way round it. And by the time we've found a way round it, things have moved on again, and it doesn't work. So that's very frustrating. And a bit… not depressing… soul destroying, because we're always playing catch up. (Person with MND, P24)

Sometimes physical symptoms necessitated *changes in many areas of life.* This included changes to their home, work, social and leisure activities. Both pwMND and caregivers spoke about the effort involved in either living with MND or looking after someone with MND. This sometimes affected thoughts and mood negatively including feelings of frustration, anxiety, tiredness or strain/burden. For example, C03, a caregiver, recalled being upset and needing someone to talk to due to exhaustion from physically moving her spouse; 'I used to get so tired that sometimes when he had gone to bed in the evening I did just feel like standing there crying.'

### Feeling unsupported by professionals or the care system

Participants reported feeling distressed when healthcare professionals or the care system did not support their needs. *Infrequent or inefficient care provision* left people feeling abandoned and left on their own to deal with MND.

> I felt completely unsupported by the neurologist or the health service, we were really just left on our own, completely in the dark as to what was gonna happen… So, it was a very, very distressing time. (Person with MND, P14)

Poor coordination between different professionals/ organisations, difficulty accessing equipment and financial benefits also triggered feelings of frustration, injustice and being 'let down' by the system.

*Poor or insensitive communication from healthcare professionals* added to emotional distress. In particular, participants described upset as a consequence of a lack of empathy from professionals, or insensitive or unsupportive presentation of care.

> Our life was shattered in 2 minutes. And I felt that was cruel, very very cruel [pause]. There should be a way of discussing it with you and saying 'well look, you have such and such a condition, but hey, this is what we can do.' Not just you've got this and throw you out, and so we had to find our own way round this problem. (Caregiver, C01)

### Strategies used to improve emotional well-being

Participants described strategies they used to improve emotional well-being (themes are numbered; subthemes in italics).

### Finding hope and positivity

Despite the many losses, participants actively found ways to introduce hope and positivity. For many people, acceptance of MND was a gradual, learnt process and even once there was acceptance, an attitude of not giving up on life was important for coping (*acceptance and not giving up*).

> I always wanna think I've got options, got somewhere to go because think at the point where you think well I've got nowhere left to go with this, that's when you may sort of deteriorate and let it get the best of you. But I think if you if you say right well I'm going to, you know, there's this trial, this might happen, I'll go and see a physio… and just see if that helps me. (Person with MND, P18)

For some, not giving up included hoping for a cure or trying alternative therapies. For others, not giving up included finding solutions to make life easier, and not letting MND define them as they continued normal and valued activities. This strategy was endorsed mainly by pwMND.

> not giving up, not allowing it to put you away in a corner somewhere, waiting for the inevitable… not allowing that to define what you do with whatever time you have left. (Person with MND, P06)

Despite severe losses of function, many participants continued *doing activities that they enjoyed and that were meaningful.* Planning ahead and having things to look forward to in the immediate future added to feelings of hope.

> Key thing is to have lots of little goals, lots of little hopes of things to do, normalities, weekends away, holidays but not in the far future. Everything's in a couple of weeks, 2–3 weeks, something to look forward to… it keeps the mind occupied to know that there is something coming up. (Person with MND, P10)

Most participants tried to *focus on the positive aspects of life.* This strategy became difficult in the context of increasing loss, but participants still stressed that it was important.

> I'm very much sort of an optimist. You can't…you know, because it's horrible we know that. But we just make [name of pwMND] as comfortable as possible. You've got to be positive, otherwise you just sink and go under. (Caregiver, C07)

To stay positive and deal with negative emotions, participants tried to reframe difficult situations by focusing on what they still can do, using humour, feeling grateful for symptoms they did not have or feeling lucky because of their circumstances or because of the support they received.

## Exerting some control

When MND took some control away, participants found other ways to exert control over their lives. Although a sense of control was important for most participants, they exerted this control in very different ways. One way was by *focusing on the present moment* and making the most of what they could currently do. This strategy was adopted mainly by pwMND; it stopped them from worrying about the future or feeling sad for what they could no longer do, and instead had a more positive impact on emotions.

> I realised there's absolutely no point in dwelling on the past because there's absolutely nothing I can do to get that back. So, it's waste of energy and there's no point in worrying about the future because there's absolutely nothing I can do about it. So, what I kind of taught myself to do is try to live in here and now because I do have some control over that. (Person with MND, P04)

Some participants described how it was helpful to exert *control over decisions about their care*, organising or planning care and taking a more active problem-solving approach. This gave people a sense of empowerment when they would otherwise be struggling to cope, and also helped people hold onto their independence or identity despite the loss.

> I hate the fact that I can't talk. Although I can talk, but it's a bit drained and doesn't sound like me. But I've done voice banking and I think you've got to be a step ahead. (Person with MND, P19)

> 'they're [professionals] pretty good at picking up what might be the next thing that's necessary or how tired I'm looking and dropping in a suggestion here and there about what you might change, what you might need next… in terms of future proofing, so you're on top of a problem and not floundering about solving it.' (Caregiver, C05)

However, some people exerted control by avoiding interactions or information that might remind them of future symptoms or deterioration and associated negative emotions.

> Everyone's different, aren't they? And their MND is different. Some people like to know the ins and out, but I don't know, I like to bury my head in the sand. Because I know that I would sit and worry about it all the time. And I don't want to. (Person with MND, P24)

## Being kinder to oneself

This strategy was mentioned by both pwMND and caregivers but for different kinds of tasks. Living with MND was often described as overwhelming and participants expressed the need for *taking a break* from MND. Caregivers spoke about the need to take a physical break from MND care; while pwMND spoke about taking a break from MND by doing regular activities and not thinking about MND.

> I'll have days where I deliberately avoid looking at anything and trying just to have days or a couple of days where I don't think about MND. (Person with MND, P18)

Some participants also expressed the need to give themselves *time and space to adjust to changes* in symptoms. This was done by allowing themselves space to vent if they felt overwhelmed, pacing their activities, lowering expectations, modifying how tasks were done, or asking for help.

> Keep doing the things you enjoy, just lower your expectations and standards so you achieve and don't feel disappointed. (Person with MND, P21)

## Experiencing support from others

*Emotional and practical support* from healthcare professionals, friends and family helped pwMND and caregivers cope with any emotional distress. Emotional support included being empathic, listening, being encouraging and positive. Participants also valued support from other pwMND because they felt truly understood and less alone.

> I had a confirmation diagnosis at [hospital] and then I had a phone call from the [hospice] asking if I wanted to attend a kind of MND first contact group… and it's been a fantastic thing and I still go now. And that was really good, really supportive. (Person with MND, P19)

There were some differing views about the timing of offering support, especially just after diagnosis. Some participants wanted access to information and support almost immediately, however others needed some time to adjust before they accessed support.

*Supportive communication from healthcare professionals and reliability of care* helped patients and families feel reassured and confident about their care, which led to positive experiences and emotions. This included providing information based on the patient and families' readiness, communicating information in a sensitive and empathic manner, focusing on what can be done in terms of care and timely provision of care and equipment.

> I would also mention my occupational therapist who has been brilliant at assessing my needs and getting in equipment quickly, usually just before they were needed. This has given me more confidence in the care and support I am given. (Person with MND, P22)

## DISCUSSION

This study advances our understanding of what influences emotional distress and well-being among pwMND and their caregivers. Our findings come from a diverse sample, thereby highlighting key triggers of distress and coping strategies used by people with different abilities, symptoms and at different stages of having MND. PwMND and caregivers use coping strategies differently but still rely on similar concepts (hope, positivity, control, self-kindness, social support) to improve emotional well-being.

Some of our findings are in line with previous MND research, such as the distress caused by loss of ability and threats to the future.[10 26 27 37] Previous research has also identified distress caused through not receiving appropriate professional support.[6 7 10 31 38 39] We highlight the emotional distress triggered by multiple and constant changes brought about by disease progression and constantly 'playing catch up.' This is a new finding in terms of emotional distress for pwMND, but has been described in the literature around caregiver experience.[17 40 41] This is an important finding for intervention development, and we need to ensure that interventions are perceived as manageable, not burdensome.

In terms of emotional well-being, hope and control are particularly important for MND.[10 37 39 42 43] Soundy and Condon have developed a model to show how concepts of hope and control can affect mental well-being in MND.[10] Hope and control can be difficult concepts to apply in MND where hope and control are constantly being threatened. Our findings highlight these complexities, and provide examples of how pwMND and caregivers use hope and control to maintain and improve well-being, despite this threat. For example, we found that many people use meaning-based coping strategies by either reappraising difficult circumstances in a positive way or doing activities that were important and meaningful. However, for some accepting their circumstances was difficult and hope meant adopting an attitude of not giving up and not letting MND define them. Psychological interventions such as acceptance and commitment therapy can be useful and can foster meaning-based coping strategies.[44–46] However, there might be certain challenges in how interventions are presented to people who may find acceptance difficult with MND. Our findings also show how control is exerted differently in the face of loss, by focusing more on the present moment, or by making decisions about receiving information and care. This has implications for support services, especially in providing options for care, equipment and communication aids.[47] The importance of focusing on the present demonstrates the value of mindfulness-based approaches for psychological interventions. The differences in how people use control highlights the importance of having flexible support that is tailored to the individual's needs and coping preferences.[46] Another novel finding in relation to emotional well-being and MND is the importance of self-kindness and self-care. Self-compassion has been associated with adaptive coping strategies and well-being in other chronic illnesses[48–50] and similar approaches may be useful for pwMND and their caregivers.

### Strengths and limitations

We aimed to recruit a diverse range of participants and succeeded in sampling people with different symptoms of varying severities, particularly people with difficulties speaking. We did not manage to recruit many people with mild cognitive impairment and could have benefitted from a more targeted sampling approach. Although we captured the experiences of newly diagnosed people and people who had MND for several years, we could not ascertain if we captured the experiences of people who were at the end stages. Conducting email interviews ensured we included the experiences of people with speech difficulties; however, there were some methodological difficulties in using prompts and asking follow-up questions. Despite these limitations, including the experiences of underrepresented groups gives us confidence in the application of these findings for people with different symptoms and at different stages of MND.

### CONCLUSION

We provide a broad understanding of what impacts emotional distress and well-being among pwMND and their caregivers. Findings have important implications for psychological interventions, services and professionals that support pwMND and their families. Any communication and support provided for pwMND and their caregivers, needs to pay attention to concepts of hope, control and compassion, and how individuals may use these concepts differently to cope with the emotional impact of MND.

**Contributors** CP: main contributor to the design of the study, ethics application, recruitment and data collection, data analysis, reporting and publication of findings. AWAG: involved in obtaining funding for the study, contributed to the study design, supervised CP during recruitment, data collection and data analysis and contributed to the reporting and publication of findings. LY: involved in obtaining funding for the study, contributed to the study design and the reporting and publication of findings. LD: involved in obtaining funding for the study, contributed to the study design, supervised CP during recruitment, data collection and data analysis, and contributed to the reporting and publication of findings.

**Funding** This work was supported by the Motor Neurone Disease Association, UK under Grant (Yardley/May17/891-792).

**Competing interests** None declared.

**Patient consent for publication** Not required.

**Ethics approval** We obtained ethics from the University of Southampton ethics committee (ERGOII-46996).

**Provenance and peer review** Not commissioned; externally peer reviewed.

**Data availability statement** Data are available upon reasonable request. Anonymised interview transcripts can be made available upon reasonable request from the corresponding author Cathryn Pinto (C.L.Pinto@soton.ac.uk).

**ORCID iDs**
Cathryn Pinto http://orcid.org/0000-0001-7607-7192
Adam W A Geraghty http://orcid.org/0000-0001-7984-8351
Laura Dennison http://orcid.org/0000-0003-0122-6610

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
