## [Reviewer comments · BMJ Open]

ARTICLE DETAILS

TITLE (PROVISIONAL)	Emotional distress and wellbeing among people with Motor Neurone Disease (MND) and their family caregivers: a qualitative interview study
AUTHORS	Pinto, Cathryn; Geraghty, Adam; Yardley, Lucy; Dennison, Laura

VERSION 1 – REVIEW

REVIEWER	Soundy, Andrew University of Birmingham
REVIEW RETURNED	06-Oct-2020

GENERAL COMMENTS	This is an important topic. I have given my considerations below. I would be interested to see other reviewers reports and considerations. Introduction I think the danger of the topic given the results it is attempting to cover a lot of aground and terms or concepts and I am not sure exactly how it furthers past knowledge for me there is more value in certain areas (I have noted below). Essentially I would like a stronger rationale for what is not known currently Definition of emotional distress – would help there reader Methods Design cant be describe as a qualitative study – page 15. Needs a methodology and paradigmatic stance Should demographics be within the results section? I would have a clear eligibility sub-title for both groups you are focusing on Was the interview piloted? How were the questions decided upon? Can you have a section on trustworthiness and link this to your paradigmatic framework? Can you have a section for justifying your sample size? Results Focal area: • Reduced autonomy• Patients have anger, sadness or grief – caregivers just have sadness?• Changes in identity – you mean social identity?• Changes in relationships and become an burden• Practical concerns over managing tasks
--

	 • Threatened future • Why threatened future – linked to interactions – VALUE for me is here • Uncertainty of disease progression • Nature of changing symptoms • Changes in areas of life • Feeling unsupported by HCPS – VALUE for me is here • Hope and acceptance • Maintaining activities • Focus on positive aspects of life • Exerting control • Kinder to ones self • Time and space to adjust • Giving support – emotional and practice – VALUE for me is here Discussion as introduction
--	--

REVIEWER	Galvin, Miriam University of Dublin Trinity College, Academic Unit of Neurology, Trinity Biomedical Sciences Institute
REVIEW RETURNED	16-Oct-2020

GENERAL COMMENTS	These authors aimed to understand experiences of people with MND and their caregivers, the impact of the living with condition on their emotions and wellbeing. Their qualitative study used semi-structured interviews to collect this information from people with MND and caregivers in the UK (n=35). The authors suggest the need to focus on concepts of hope, control and compassion and they discuss implications for psychological interventions. PPI involvement made clear Of interest to readers Minor revisions suggested Specific points to address: According to the authors, purposive sampling was used for recruitment, yet there was underrepresentation of mild cognitive impairment, would further targeted sampling have been used to address that purposively? Could the authors comment on any differences noticed in interviewing in person, phone, email etc? and was there any difference in the findings by mode of interview, if so what were they? Methods: Were there any inclusion and exclusion criteria for the caregiver participants? How was the capacity to consent assessed? How was informed consent obtained if interviews were not in person? How did the phenomenological orientation (Coreq 9.) inform theme development? Participant checking is mentioned in checklist but not discussed again Analysis and Discussion
--

	The illustrated quotes are from people with MND, suggest including some from carers especially in the section Strategies used to improve emotional wellbeing Was there much divergence or convergence in the themes and subthemes among PwMND and carers?
--	--

VERSION 1 – AUTHOR RESPONSE

Reviewer 1:

1. I think the danger of the topic given the results it is attempting to cover a lot of ground and terms or concepts and I am not sure exactly how it furthers past knowledge for me there is more value in certain areas (I have noted below). Essentially I would like a stronger rationale for what is not known currently

Response: We agree there is a lot to cover in this paper about emotional distress and wellbeing. We feel it is important to report the less novel results alongside the more novel results, because it is important to show the full picture of emotional distress and wellbeing, especially since we have included experiences of people who may have been under-represented in previous research (e.g. people with speech difficulties and those very recently diagnosed).

We have now made substantial edits to the introduction, results and discussion section to explain the more novel aspects of our findings. We have made changes to the introduction (Pgs 4-6) to make the rationale stronger. We have explained the specific changes to the results and discussion sections, as they appear in this points below.

2. Definition of emotional distress – would help there reader

Response: Thank you, we have added this to clarify that with the term emotional distress we are referring to the broad range of negative emotional states, beyond anxiety and depression. (Pg 4)

3. Design can't be describe as a qualitative study – page 15. Needs a methodology and paradigmatic stance

Response: We have added more details about the study design on Pg 7. It now reads 'Qualitative study using in-depth semi-structured interviews and reflexive thematic analysis, in line with an interpretivist approach.'

4. Should demographics be within the results section?

Response: We have added the demographic details and Table 1 to the results section and edited the methods section accordingly. Details are now on Pgs 12-13

5. I would have a clear eligibility sub-title for both groups you are focusing on

Response: Thank you. We have added the subheading on Pg 7

6. Was the interview piloted?

Response: Yes, we piloted the interview topic guide with the PPI group members (mentioned on Pg 11)

7. How were the questions decided upon?

Response: We have added details about the development of the interview questions (Pg 10) 'The interview topic guide was developed iteratively by CP, LD, AG and patient and public involvement members. In line with an interpretivist approach questions were broad and open-ended; follow-up questions were led by participants' responses. The final interview topic guide (Supplementary file 1)

covered questions about people's experiences living with MND, with a focus on their thoughts and feelings and coping with emotional concerns.'

8. Can you have a section on trustworthiness and link this to your paradigmatic framework?

Response: We agree that it is important to demonstrate the trustworthiness of our study and methods, and how this links to our paradigmatic framework. Instead of a separate section, we have chosen to integrate this information throughout the paper. We have included information on the trustworthiness of our study based on quality criteria by Yardley (2000) (Reference: Yardley L. Dilemmas in qualitative health research. *Psychology and health*. 2000 Mar 1;15(2):215-28) Please see the following points:

Sensitivity to context: In the introduction we explain the existing literature and findings on emotions, we query the need for a more in-depth focus on the emotions experienced in relation to MND (Pgs 4-5). We are also transparent about the background and training of the researchers (Pg 9, line 18) and acknowledge that this is different from the sample and have therefore, included PPI contributors at several stages of the research to ensure methods are sensitive to the context of those being studied (Pg 11).

Commitment and rigour: We have included PPI contributors from the initial stages of designing the study methods to the interpretation of findings (Pg 11) and have kept participants informed about the study results (Pg 11, line 5-6). Our analysis and findings strike a balance between capturing the breadth of experience of pwMND and caregivers, and providing an in-depth account of new themes, where we present both convergent and divergent cases (Results section).

Transparency and coherence: We have been transparent about methods for data collection and analysis throughout, for example on Pgs 9-10 we detail our recruitment methods, we have added more details about the development of interview question (Pg 10) and have detailed the steps taken and the people involved at each step of data analysis (Pgs 10-11). In terms of coherence, we have stated our paradigmatic framework (Pg 7, line 4), and how this fit in with our methods for data collection (Pg 10, lines 4-6) and analysis (Pg 10, lines 16-19).

Impact and importance: In the introduction, we demonstrate that this study began with the aim of creating impact and providing relevant information to aid intervention development (Pg 5). We also discuss how our findings can be used to inform psychological interventions and MND services more generally (Pgs 26-27).

9. Can you have a section for justifying your sample size?

Response: We have now added this information on Pg 7 - We aimed to recruit 20-30 pwMND and used purposive sampling to represent people with difficulties with movement, speech and cognition, and different lengths of time since diagnosis. Caregiver participants had fewer sampling criteria (age, gender), therefore we aimed to recruit 10-15 caregivers.

10. Focal area:

- Reduced autonomy
- Patients have anger, sadness or grief – caregivers just have sadness?
- Changes in identity – you mean social identity?
- Changes in relationships and become an burden
- Practical concerns over managing tasks
- Threatened future
- Why threatened future – linked to interactions – VALUE for me is here
- Uncertainty of disease progression
- Nature of changing symptoms
- Changes in areas of life

- Feeling unsupported by HCPS – VALUE for me is here
- Hope and acceptance
- Maintaining activities
- Focus on positive aspects of life
- Exerting control
- Kinder to ones self
- Time and space to adjust
- Giving support – emotional and practice – VALUE for me is here

Response: We have noted the areas where the reviewer finds value, we agree with some of the comments, but we feel that there are some themes that have already been discussed well in previous literature (losing function and ability and having a threatened future). We decided to focus on the novel theme of 'keeping up with multiple and constant changes' and 'being kinder to oneself', we have highlighted this in the discussion section. We have added more description and quotes in the results section to further explain the themes 'feeling unsupported' and 'experiencing support from others' (Pgs 18, 24 and 25). We also agree that we could be more specific about how hope and control link to wellbeing in the context of MND, and have provided some additional quotes to illustrate this (Pgs 19-22).

There were a few clarifications - In response to the second bullet point, we agree this was not clear and have clarified that these emotions were similar for caregivers (Pg 14). In response to the third bullet point, we have specified that we meant self-identity on Pg 15.

11. Discussion as introduction

Response: We have made the following changes in the discussion section:

1. Altered the language so that it is more clear what is known and what is novel (Pg 26)
2. Elaborated on our discussion of hope and positivity and exerting control and the particular relevance of this to MND (Pgs 26-27)

Reviewer 2:

1. These authors aimed to understand experiences of people with MND and their caregivers, the impact of the living with condition on their emotions and wellbeing. Their qualitative study used semi-structured interviews to collect this information from people with MND and caregivers in the UK (n=35). The authors suggest the need to focus on concepts of hope, control and compassion and they discuss implications for psychological interventions. PPI involvement made clear

Of interest to readers

Minor revisions suggested

Response: Thank you for your feedback

2. According to the authors, purposive sampling was used for recruitment, yet there was under representation of mild cognitive impairment, would further targeted sampling have been used to address that purposively?

Response: Yes, we agree a more targeted approach would have been helpful. We have included this suggestion this in the discussion (Pg 28)

3. Could the authors comment on any differences noticed in interviewing in person, phone, email etc? and was there any difference in the findings by mode of interview, if so what were they?

Response: In terms of the methods, there wasn't any difference between the face-to-face and phone interviews, email interviews had less thick data. Follow up questions were also more difficult via email. We have added this information in the article summary, strengths and limitations (Pg 3). For the analysis, there weren't any differences in the findings based on mode of interview, so we haven't

included this information in the paper.

4. Were there any inclusion and exclusion criteria for the caregiver participants?

Response: We included caregivers above 18 years of age, both current and recently bereaved (Pg 7). No further inclusion criteria

5. How was the capacity to consent assessed?

Response: We have added these details on Pg 7. 'Participants were above 18 years of age, had an MND diagnosis, and had mental capacity to consider participation in the study (assessed by the researcher through correspondence about the study).'

6. How was informed consent obtained if interviews were not in person?

Response: Thank you for highlighting this, we have added this information on Pg 9 - Before each interview, participants gave written informed consent and filled a demographic/clinical details form. This process was completed either in-person, by post or email, based on the interview mode.

7. How did the phenomenological orientation (Coreq 9.) inform theme development?

Response: The label we used was incorrect and we have now corrected this to say an interpretivist approach. On Pg 10, we explain how this influenced theme development - In line with an interpretivist approach, we used an inductive approach to data analysis, and included convergent and divergent cases in theme development.

8. Participant checking is mentioned in checklist but not discussed again Analysis and Discussion

Response: This was not really participant checking, we have amended this in the COREQ checklist and added a sentence to the paper to reflect that we meant a lay summary of the findings was sent to participants (Pg 11).

9. The illustrated quotes are from people with MND, suggest including some from carers especially in the section Strategies used to improve emotional wellbeing Was there much divergence or convergence in the themes and subthemes among PwMND and carers?

Response: This is an important point, sorry to have overlooked it. We have now added more caregiver quotes in the strategies section (Pgs 21 to 25). Overall, there was convergence in the themes among PwMND and carers. There was some divergence in the subthemes; these have been highlighted on Pg 20 (acceptance and not giving up), Pg 21 (focusing on the present moment), these particular strategies were endorsed by pwmnd, not caregiver participants.

VERSION 2 – REVIEW

REVIEWER	Soundy, Andrew University of Birmingham
REVIEW RETURNED	29-Dec-2020

GENERAL COMMENTS	General I think the methods have vastly improved and the authors have been responsive to this and to other points. I can also see improvements in structure. However, two critical problems for me are: (1) I am unsure about the focus and terms used and definition of these terms – or maybe how they are conveyed. (2) I am not convinced by the results still (see below). I understand the editor can take a view on this. Either way I believe it is time time for me to take a side step on this article. I wish you the best and want to encourage you with further work in this
--

important area. Please note my comments are only intended to be an honest reflection of my concerns and I understand that others may not share these concerns and you may also take a different view - that is part of academia.

I would like to explain my considerations:

1. Focus on emotional distress

Your definition of emotional distress and using its focus doesn't make sense to me as I think there is more to explain e.g., I would define negative emotional states as sadness, anxiety, resentment, frustration, depression – so there is a focus on them being unpleasant and then differentiated by energy. I wouldn't put these emotions with hope and hopelessness and demoralisation as I think it isn't that simple (from reading hope theory etc). As a result I found it hard to work out the focus.

Before you define emotional distress you talk about the factors that affect it e.g. low self-esteem, end of life concern etc – so I think you need to move this from line 9 to the end of line 6 before the factors that influence them to help the reader understand what you are referring to.

2. Problems with the results

I won't go over my comments on the results again. But I just want to make an illustration of what I perceived to be the problems from the last results and from two reviews I have co-authored

references
<https://pubmed.ncbi.nlm.nih.gov/28931454/> - caregiver review not referenced

<https://www.ncbi.nlm.nih.gov/pmc/articles/PMC4428059/> - your article reference 10

1.

I can see that your section on losing function or ability identifies the importance of a reduced sense of autonomy and control – the results from reference 10 has a section on this entitled Loss of control, agency, and autonomy within theme 3 of review and the importance of autonomy, control and agency in theme 4. Within theme 1: 2 subthemes cover loss and change to social and occupational relationships and then physical and functional losses. The problem for me is that reference 10 alone shows that multiple qualitative studies have considered these aspects in detail – thus for me the introduction needs to tell the reader what is different here as I can't work it out, for me the past qualitative work has this covered and the review gives greater detail from multiple qualitative results.

Another example could be your theme on feeling unsupported by professional or the system, again reference 10 has sections showing multiple studies already have considered interactions, isolation and relationships, problems with emotional support. Alternatively your theme around finding hope – which seems covered by theme 5 from reference 10. The reader needs to know how it is different for instance compared to this.

I understand you want to present an overview of the findings but at present I can't see what the contribution to knowledge is and I need this pointed out more. Given that these references from myself are

	a few years out of date and other reviews are likely available which need consideration too. I hope that makes sense. I wish you the best with your work.
--	---

VERSION 2 – AUTHOR RESPONSE

Reviewer 2 appears to have accepted the paper following their original suggestions for minor edits.

Reviewer 1 had comments on:

- 1) How emotional distress is conceptualised.
- 2) Novelty about the findings and the importance of the contribution to the literature.

The conceptualisation of emotional distress comment has been easily addressed with minor edits to the manuscript introduction. We have also responded to the novelty/contribution comments with some further manuscript edits (see table of comments & responses, page numbers refer to the manuscript with tracked changes). However, in the remainder of this letter we also rebut some of the concerns.

Reviewer 1 has conducted two useful meta-syntheses and these have been discussed in our paper. We have checked that there are no other more recent reviews in the literature on this topic that haven't already been cited in our paper. When we began this research, we were familiar with the existing literature and proceeded in two novel ways. Firstly we focused our research question on emotional distress and wellbeing and asked people directly about this. We considered it important to interview people with a comprehensive set of questions about wellbeing, and emotional distress as an interview focused on this may allow them to provide accounts that are different and/or more detailed than what we already know from the literature. Previous qualitative research (represented in Reviewer 1's meta-syntheses) has either explored general experiences of living with MND or focused on emotions in relation to particular aspects of MND and MND care (e.g. Locock et al, 2012; Pavey et al, 2013; Whitehead et al, 2012). Secondly, we included groups of people with MND that have typically been neglected or under-represented in previous research (i.e. people who have difficulties with speech, were very recently diagnosed (less than 6 months), or who experienced some cognitive impairment). These clinical characteristics are common and may also influence emotional experiences. Our study therefore, aimed to represent their views in the literature.

As is typical of qualitative research we adopted an exploratory approach and an inductive analysis, and could (and should) not have anticipated the findings ahead of time. We accept Reviewer 1's conclusion that there is similarity between our findings and what is already described in the literature. We acknowledge that we may not have communicated the overlap and novelty of our findings strongly enough originally and the reviewer's comments have prompted us to strengthen this.

Nonetheless we maintain that our findings make some novel contributions, in particular the themes 'Keeping up with constant and multiple changes' and 'Being kind to oneself', as well as the examples provided about the different and complex ways participants used hope and control in the context of coping with MND. The remaining themes are indeed similar to findings from previous research. However, by including under-represented groups and still arriving at similar findings gives us confidence in the literature and its transferability to a more representative population of people with MND. It is important scientifically to publish confirmatory research. We chose to submit to BMJ Open on the basis that it would be open to publishing findings of this nature given the editorial policies, scope statement and guidance for reviewers on the BMJ Open website:

- "Editorial decisions will not judge articles for importance, relevance or originality".
- "studies that may be judged unoriginal by other journals because they replicate in different settings work that has already been done elsewhere. It can be important to clinical practice or health policy to replicate evidence that has already been established in one type of setting (for example, in well-resourced healthcare settings)"
- "studies by young and new researchers. We recognise that researchers want and need to publish their work while they are learning and developing their ideas and skills: BMJ Open is keen to encourage you to do so with full transparency and cautious interpretation" (Lead Author is a PhD student)
- "Reviewers will not be asked to judge importance or breadth of appeal. Readers will be able to make these judgements for themselves"
- "(Instructions for reviewers) We do not need you to comment on the work's importance to general readers. Please consider it for scientific reliability and ethical conduct".

We believe that Reviewer 1's latest comments relate to his perceptions of the originality and importance of the findings, rather than the rigour and quality of the research conducted and that this is not in line with BMJ Open policies.

We still maintain that our paper makes an important contribution to the field, particularly in terms of relating our findings to psychological intervention development. In terms of clinical relevance, there have been recent calls for more interventions to support the psychological needs of people with MND (British Psychological Society, 2021). Our paper is timely and furthers this agenda by discussing how constructs such as hope, control, and compassion can be used to make interventions acceptable and engaging for people with MND and family members.

We are looking forward to hearing your decision about how to proceed. It seems that Reviewer 1 has now disengaged from the peer review process (“I understand the editor can take a view on this. Either way I believe it is time for me to take a side step on this article I wish you the best and want to encourage you with further work in this important area. Please note my comments are only intended to be an honest reflection of my concerns and I understand that others may not share these concerns and you may also take a different view - that is part of academia”). We would like to request that any decision is communicated to us as soon as possible, as we originally submitted this manuscript in September 2020, and following reviewer’s comments revised and re-submitted in December 2020. Knowing the outcome sooner will help us make alternative plans for publication.

Best wishes,

Cathryn Pinto

(on behalf of all the authors)

References

British Psychological Society (2021). Psychological interventions for people with Huntington’s disease, Parkinson’s disease, motor neurone disease, and multiple sclerosis: evidence-based guidance. Leicester: Author

Locock L, Mazanderani F, Powell J. Metaphoric language and the articulation of emotions by people affected by motor neurone disease. *Chronic illness*. 2012;8(3):201-13.

Pavey A, Allen-Collinson J, Pavey T. The lived experience of diagnosis delivery in motor neurone disease: a sociological-phenomenological study. *Sociological research online*. 2013;18(2):36-47

Whitehead B, O’Brien MR, Jack BA, Mitchell D. Experiences of dying, death and bereavement in motor neurone disease: a qualitative study. *Palliative Medicine*. 2012;26(4):368-78.

Reviewer’s comments	Response
(1) I am unsure about the focus and terms used and definition of these terms – or maybe how they are conveyed. Focus on emotional distress:	We have made a couple of edits to the introduction in order to clarify our definition and focus. 1. We begin the introduction by talking broadly about psychological impact. And then discuss how emotional distress and wellbeing is conceptualised in the

You definition of emotional distress and using its focus doesn't make sense to me as I think there is more to explain e.g., I would define negative emotional states as sadness, anxiety, resentment, frustration, depression – so there is a focus on them being unpleasant and then differentiated by energy. I wouldn't put these emotions with hope and hopelessness and demoralisation as I think it isn't that simple (from reading hope theory etc). As a result I found it hard to work out the focus. Before you define emotional distress you talk about the factors that affect it e.g. low self-esteem, end of life concern etc – so I think you need to move this from line 9 to the end of line 6 before the factors that influence them to help the reader understand what you are referring to.	literature and what we are referring to specifically in this paper (Pg 4, lines 17-22). We have kept our definition quite broad in keeping with our methodological stance of being led by how participants described the emotions they experienced and related experiences of distress and wellbeing.  2. We agree with the comment about the placement of the definition of emotional distress before talking about factors that influence it. The definition is now on Pg 4, lines 19-22, and the sentence about factors has now been moved to a subsequent paragraph (Pg 5, lines 3-5) 3. We have made some edits and added further explanation to clarify the focus of this paper. (Pg 5, lines 5-8, lines 13-14, lines 18-19)
2. Problems with the results I won't go over my comments on the results again. But I just want to make an illustration of what I perceived to be the problems from the last results and from two reviews I have co-authored references. I can see that your section on losing function or ability identifies the importance of a reduce sense of autonomy and control – the results from reference 10 has a section on this entitled Loss of control, agency, and autonomy within theme 3 of review and the importance of autonomy, control and agency in theme 4. Within theme 1: 2 subthemes cover loss and change to social and occupational relationships and then physical and functional losses. the problem for me is that reference 10 alone shows that multiple qualitative studies have considered these aspects in detail – thus for me the introduction needs to tell the reader what is different here as I can't work it out, for me the past qualitative work has this covered and the review gives greater detail from multiple qualitative results.	We acknowledge that there is some similarity between our findings and that of previous research, including the 2 meta-syntheses identified. We could not have anticipated this ahead of conducting the study, and believe that having results that confirm what has previously been found can also be helpful in increasing confidence in the literature. Having said that, we also feel that our study highlights some novel aspects. We have outlined our response to the specific feedback below:  1. We agree that the theme about losing function and ability and the corresponding subthemes have been discussed in previous literature. We have presented these results as part of an overview of the triggers of distress, as omitting it would give an incomplete picture, neglecting aspects that our participants were highlighting as important. However, in our discussion, we clearly state that this supports previous findings, citing the reviewer's meta-synthesis (Pg 23, lines 10-11)

Another example could be your theme on feeling unsupported by professional or the system, again reference 10 has sections showing multiple studies already have consider interactions, isolation and relationships, problems with emotional support. Alternatively your theme around finding hope – which seems covered by theme 5 from reference 10. The reader needs to know how it is different for instance compared to this.

I understand you want to present an overview of the findings but at present I can't see what the contribution to knowledge is and I need this pointed out more. Given that these references from myself are a few years out of date and other reviews are likely available which need consideration too.

2. In terms of the theme around being unsupported by professionals, we have mentioned in the discussion that this also echoes what has been previously found. We have now added the reviewer's citations as well to strengthen this point (Pg 23, line 12).
3. The importance of hope and control and their relevance for coping with MND has been explored in the literature. We have added some more information in the discussion to elaborate on our findings about hope and control. We are not stating that these are new concepts that haven't been previously identified or theorised, but in our paper we highlight some of the complexities when using these concepts. For example, the difficulty with hope and acceptance of the disease, or how exerting control in terms of thinking about symptoms or receiving information about the disease can differ between patients, or between patients and caregivers. These differences in how people cope is important to highlight particularly from an intervention development point of view (explained on Pg 24, lines 1-16).
4. Overall, we now more clearly acknowledge the overlap with our findings and previous research. There are also some novel aspects around coping with multiple and constant changes and around self-compassion, which have been highlighted in the discussion (Pg 23, lines 13-17, and Pg 24, lines 16-19). Having similar findings after including groups that have been under-represented in the literature, also strengthens our confidence that we have captured their experiences thoroughly as well. This has now been stated more clearly on Pg 25, lines 6-8. Given this overlap in findings, we have made our contribution clearer through edits in the framing of key findings in the discussion (Pg 23, lines 2-3), and conclusion (Pg 25, line 10, and lines 14-16).